# Improvement of Advanced Parkinson’s Disease Manifestations with Deep Brain Stimulation of the Subthalamic Nucleus: A Single Institution Experience

**DOI:** 10.3390/brainsci6040058

**Published:** 2016-12-13

**Authors:** Ahmed Rabie, Leo Verhagen Metman, Mazen Fakhry, Ayman Youssef Ezeldin Eassa, Wael Fouad, Ahmed Shakal, Konstantin V. Slavin

**Affiliations:** 1Department of Neurosurgery, University of Illinois at Chicago, Chicago, IL 60612, USA; dr_a_rabie@hotmail.com; 2Department of Neurosurgery, Alexandria University, Alexandria, Egypt; mazenfakhry56@yahoo.com (M.F.); waelfouad_67@hotmail.com (W.F.); 3Department of Neurological Sciences, Rush University Medical Center, Chicago, IL 60612, USA; lverhage@rush.edu; 4Department of Neurology, Alexandria University, Alexandria, Egypt; af_eassa@yahoo.com; 5Department of Neurosurgery, Tanta University, Tanta, Egypt; ahmedshakal@yahoo.com

**Keywords:** subthalamic nucleus, deep brain stimulation, Parkinson’s disease, neuromodulation, clinical outcome

## Abstract

We present our experience at the University of Illinois at Chicago (UIC) in deep brain stimulation (DBS) of the subthalamic nucleus (STN), describing our surgical technique, and reporting our clinical results, and morbidities. Twenty patients with advanced Parkinson’s disease (PD) who underwent bilateral STN-DBS were studied. Patients were assessed preoperatively and followed up for one year using the Unified Parkinson’s Disease Rating Scale (UPDRS) in “on” and “off” medication and “on” and “off” stimulation conditions. At one-year follow-up, we calculated significant improvement in all the motor aspects of PD (UPDRS III) and in activities of daily living (UPDRS II) in the “off” medication state. The “off” medication UPDRS improved by 49.3%, tremors improved by 81.6%, rigidity improved by 50.0%, and bradykinesia improved by 39.3%. The “off” medication UPDRS II scores improved by 73.8%. The Levodopa equivalent daily dose was reduced by 54.1%. The UPDRS IVa score (dyskinesia) was reduced by 65.1%. The UPDRS IVb score (motor fluctuation) was reduced by 48.6%. Deep brain stimulation of the STN improves the cardinal motor manifestations of the idiopathic PD. It also improves activities of daily living, and reduces medication-induced complications.

## 1. Introduction

The deep brain stimulation (DBS) system consists of a lead that is implanted into a specific deep brain target. The lead is connected to an implantable pulse generator (IPG), which is the power source of the system. The lead and the IPG are connected by an extension wire that is tunneled under the skin between both of them. This system is used to chronically stimulate the deep brain target by delivering a high-frequency current to this target [1,2].

James Parkinson was the first to describe Parkinson’s disease (PD) in 1817; he described it as a combination of tremor, rigidity, postural abnormalities, and bradykinesia [3]. The main step that marked the onset of stereotactic surgery and the surgical treatment of different movement disorders was in 1947, when Ernest Spiegel and Henry Wycis invented the first frame-based stereotactic apparatus “stereoencephalotome”. This was the first device to be used for localization of targets in the living human brain [4]. They performed the first stereotactic thalamotomy and pallidotomy, but the clinical effects were disappointing [5,6].

The credit goes to Leksell for using the posteroventral pallidum as the target for lesioning [7]. At the same time, the ventrolateral (VL) nucleus of the thalamus emerged as a target for lesioning, and with time it replaced pallidotomy for the treatment of tremors [8,9,10].

The discovery of Levodopa in the late 1960s led to a decline in surgeries for PD. Lesioning of the ventral intermediate (Vim) nucleus of the thalamus and the globus pallidus internus (GPi) continued to be the major surgical targets.

The first published work describing the use of DBS in the treatment of PD was by Benabid et al. in Montreal, France in 1987. They proved that high-frequency DBS was able to mimic, in a reversible and adjustable manner, the effects of ablation of Vim as the target to control tremors [11,12,13]. The first attempt to use the GPi as the target of DBS to treat PD was by Siegfried and Lippitz, published in 1994. DBS of the GPI proved efficiency in controlling tremors, bradykinesia, and drug-induced dyskinesias [14,15,16]. The subthalamic nucleus (STN) was investigated in animal studies as a target for Parkinson’s disease surgery [17]. Lesioning of the STN in humans proved to be effective in reducing the three cardinal symptoms of PD [18,19]. Again, Benabid and the Grenoble group were the pioneers in using DBS of the STN for the treatment of PD in 1994, based on findings from animal studies [20,21]. This led to the approval of DBS by the FDA as a method of treatment of PD. Since then the STN has been the target of choice for DBS in PD patients. It proved to be superior to medical therapy in controlling tremors, rigidity, and dyskinesia in advanced stages of PD [22,23,24,25,26].

Shakal et al. reported the first use of STN-DBS for the management of PD in Egypt in 2011 [27]. At the University of Illinois at Chicago (UIC), USA, DBS surgeries started with the work of the senior author (KVS) in early 2001. The work we are presenting here is a collaborative work between the Neurosurgery Department of Alexandria, Egypt and that of UIC. Here we present the STN-DBS experience at UIC, describe our surgical technique, and report our clinical results and morbidities. Our objectives are to evaluate the clinical outcome of STN-DBS in PD and share our experience in this field.

## 2. Methods

After obtaining an appropriate IRB approval, we retrospectively analyzed the data of 20 patients diagnosed with advanced PD who underwent bilateral STN-DBS at the UIC in the period from 2013 to 2014. Patients who qualified for surgery had idiopathic PD and showed sustained response to levodopa, with a minimum of 30% improvement in Unified Parkinson’s Disease Rating Scale (UPDRS) motor subscore following a levodopa challenge. Most patients had severe levodopa-related motor response despite optimal dose adjustment, and/or disabling tremors. We excluded patients with atypical Parkinsonism as multiple system atrophy (MSA), progressive supranuclear palsy (PSP), corticobasal degeneration, vascular, and drug-induced parkinsonism. We also excluded patients with severe cognitive impairment or dementia (Mattis Dementia Rating Scale <130 or Mini Mental Status examination ≤24), patients with severe uncontrolled psychiatric illness or depression (Beck Depression Inventory II score >19), and patients with magnetic resonance imaging (MRI) features of moderate to severe cortical atrophy, ventricular enlargement, and significant white matter changes or other significant intracranial lesions such as tumor, arteriovenous malformations, etc. We also excluded patients with other significant illnesses.

### 2.1. Pre-Operative Patient Assessment and Selection

The first step of the patient assessment was to confirm the diagnosis of primary PD and exclude other forms of movement disorders and atypical forms of parkinsonism. To make the diagnosis, the patient must have at least two of the three motor features (rest tremors, bradykinesia, and rigidity), and bradykinesia must be one of those two features. The patient must have a good response to dopaminergic drugs, as poor response suggests an atypical parkinsonian syndrome. We only included patients who had the disease for more than five years; this is to follow the recommendation of the core assessment program for surgical interventional therapies in PD (CAPSIT-PD) committee [28,29,30,31].

Each patient was asked to come for a second appointment after being off medications for 12 h to be evaluated for surgery. At this visit, the patient was assessed for mental state, behavior, and mood (part I of the UPDRS) [32,33,34,35]. Then the patient was assessed for activities of daily living (ADL), using part II of the UPDRS in both the “on” and “off” states. Then the patient was evaluated for levodopa-related complications using the UPDRS part IV [33,34,36], including the duration of motor fluctuation (items 36–39) and the severity of levodopa-induced dyskinesia (items 32–35). In the early phase of PD, symptoms can be controlled with dopaminergic medications [37,38]. After five or more years of dopaminergic therapy, about 50% of patients begin to experience motor fluctuations and dyskinesia and may become candidates for DBS [37,38]. In the late phase of PD, some patients become unresponsive to levodopa. Those patients are not considered surgical candidates for DBS [37,38]. Then the patient was assessed using the levodopa challenge test and the UPDRS motor scoring (part III) and video recorded. First, the scale was performed in an “off” state. Then the patient was given a supra-therapeutic dose of levodopa (1.5 times the patient’s current dose) and the UPDRS motor scale was assessed again during the “best on state”. At this visit, we also calculated the axial score by summing the motor subscores: speech, gait, posture, and postural stability (items 18, 28, 29, and 30 of the UPDRS part III). We also assessed the Modified Hoehn and Yahr Rating Scale (HYRS) [39], and the Schwab and England Rating Scale (SERS) [40]. Both of them were done in the “off” state. We also calculated the levodopa equivalent daily dose (LEDD) [41].

### 2.2. Neuropsychological and Psychiatric Evaluation

A dedicated psychologist then assessed the patient during the best “on” state.

The tests used for assessment were: Mattis Dementia Rating Scale (MDRS) [42], Beck Depression Inventory II (BDI-II) [43,44], Independent Living Scale (ILS)—Health and Safety, Mini Mental Status Exam (MMSE) [45,46], Peabody Picture Vocabulary Test—Fourth Edition (PPVT-4), Wechsler Adult Intelligence Scale for DSM-IV (WAIS-IV)—Digit Span, Wisconsin Card Sorting Test (WCST), Hopkins’ Verbal Learning Test—Revised (HVLT-R), and Frontal System Behavior Scale (FrSBe).

This assessment is a mandatory step before surgery. It helps to exclude any patient with severe cognitive and or behavioral impairments (Mattis Dementia Rating Scale <130 or Mini Mental Status examination ≤24), severe uncontrolled psychiatric illness or depression (Beck Depression Inventory II score >19). This assessment also helped to establish the baseline of the mental, verbal, and frontal lobe functions for further follow-up.

### 2.3. Surgery

The surgery was done in two stages. In the first stage, we implanted DBS electrodes under local anesthesia using the frame-based stereotactic technique. The patient was instructed to stop all anti-Parkinsonian medications 12 h before surgery to facilitate microelectrode recording (MER), and allow clinical assessment during stimulation. The first step was the application of the Leksell frame Model G (Elekta Instruments, Inc., Atlanta, GA, USA) to the patient’s head (Figure 1).

A high-resolution MRI of the patient’s brain with 3-tesla scanner (Signa 3T94 VHi; General Electric Medical Systems, Milwaukee, WI, USA) was done. Two main sequences were obtained. The first is a 3D T1-weighted, spoiled gradient echo imaging of the entire head (section thickness: 2 mm; field of view: 26 × 26 cm; TR: 7.0–8.0 ms; TE: ~400 ms; flip angle: 12; band width: 31.25 KHz; acquisition time: <7 min). The second sequence is high-resolution, contiguous, T2-weighted, fast spin-echo imaging through the region of the midbrain and basal ganglia (section thickness: 1.5 mm; slice interval: 0 mm; matrix size: 512 × 512; field of view: 26 × 26 cm; TR: 4600–6200 ms; TE: 95–108 ms; acquisition time: <5 min) (Figure 2).

At the end of the scan, we chose an axial T2 image (or two adjacent images) in which both the AC and the PC are seen (Figure 3). With simple arithmetic equations based on the Leksell frame coordinates system, we were able to calculate the stereotactic coordinates of the mid-commissural point (MCP), and the STN directly from the MRI coordinates of the AC and the PC (Figure 4). Based on the known anatomical relationship of the STN to MCP from the previous anatomical studies and stereotactic atlases [36,47,48,49,50,51,52,53,54], we selected the STN target at 12 mm lateral, 3 mm posterior, and 6 mm inferior to the MCP.

The second method we used to calculate the STN coordinates was direct visualization of the STN on a T2 weighted MRI (Figure 5) [55]. The STN is the almond-shaped hypointense structure located lateral and anterior to the red nucleus. We identified an axial T2 image that showed the largest red nuclei circumference, and then we drew a line from the midline, medial to lateral, along the anterior edge of RN. The center of the STN was chosen at the extension of this straight line about 12 mm from the midline. Then the coordinates were calculated using the same Excel worksheet. Another method of the STN coordinates localization was done in the OR, using the FrameLink software, which is a part of the StealthStation navigation system (Medtronic, Minneapolis, MN, USA) (Figure 6). The software compensates for head and frame tilt in any direction. It allows calculation of the STN coordinates and planning of suitable entry point and trajectory of the DBS electrode that avoid going through the cortical sulci, the ventricles, or any cerebral blood vessels. The final coordinates for the procedure were derived from all the previous techniques and subsequently adjusted using intraoperative electrical microrecording and macrostimulation.

In the operating room, the patient was placed on the operating table with a Leksell frame secured to the table using a Mayfield adapter. The C-arm was placed around the patient in order to use intraoperative fluoroscopy for electrode tracking and positioning (Figure 7). We used transparent sterile drapes to allow easier communication with the patient and observation of the patient’s symptoms during this awake procedure (Figure 8). Two semicircular incisions were made on both sides of the midline (Figure 9). Then we drilled two burr holes, one on each side, 1 cm anterior to the coronal suture and 2–3 cm lateral to the midline. We started the surgeries with the left side and then shifted to the right side.

We performed microelectrode recording (MER) of the brain activity using a NeuroNav microelectrode recording system (AlphaOmega, Nazareth, Israel) (Figure 10). Fluoroscopic confirmation of the target approach was obtained at 5 mm intervals, 2 mm above the target, and at the target (Figure 11).

After identification of the STN borders and depth by the MER, we started high-frequency macrostimulation. The aim of the stimulation was to confirm the optimal target, which provided adequate control of the Parkinsonian symptoms (specifically tremors), without undesirable effects from stimulation below 4 volts. Once we reached our desired target, we removed the microelectrode and replaced it with a standard four-contact (0–3) deep brain stimulation electrode (Medtronic DBS lead 3389). Generally, we placed the deepest electrode contact (0) at or just beyond the target point. We repeated the testing using this electrode in order to confirm the reproducibility of the effects. We locked the electrode in place using a Stimloc device (Medtronic, Minneapolis, MN, USA) (Figure 12). The excess of the electrode was coiled around the burr hole to create a strain relief loop (Figure 13). Then the same procedure was repeated on the right side.

The patient returned to hospital after one week for the second-stage surgery, in which the IPG was implanted in the sub-clavicular region under general anesthesia. After surgery, the IPG was interrogated. We checked the impedance of all eight contacts and programmed the pulse width, frequency, and amplitude of stimulation. By the end of the programming, we confirmed that the amplitude was set at zero and that the voltage of the battery was in the expected range.

### 2.4. Post-Operative Patient Assessment

Immediately after the first stage of surgery, all patients had a CT scan of the head to rule out hemorrhage. All of them had an MRI of the brain on the same day of surgery or the next day before discharge to confirm accurate electrode placement.

An experienced neurologist performed the first postoperative programming session one month after surgery. This interval was given to allow the brain to recover from the surgery and the micro-lesioning effect of the electrode placement. The patient was programmed to the best setting that gave the best clinical improvement at the lowest stimulation intensity and largest therapeutic range before inducing undesirable effects. At the same time, drug doses were reduced.

Comprehensive neurological evaluation was done 12 months after surgery. At this visit, each patient came after 12 h without medical treatment. The patient was reassessed for parts I, II, and IV of the UPDRS. After this, each patient was assessed using the UPDRS III and video recorded in four conditions: “on” stimulation and “off” medication; then “off” stimulation and “off” medication, after switching off the stimulation for at least 1 hour; then “off” stimulation and “on” medication, after administration of a supra-threshold dose of levodopa and waiting for the “best on state”; then “on” stimulation and “on” medication, after turning on stimulation using the chronic stimulation parameters. We also calculated the axial score, MMSE and SERS [40]. We recalculated the LEDD for the patient’s recent anti-Parkinsonism medications. We documented any surgical-, device-, or stimulation-related undesirable effects.

### 2.5. Data Collection and Statistical Analysis

The collected data were coded, tabulated, and statistically analyzed using the IBM SPSS statistics software version 22.0 (IBM Corp., Chicago, IL, 2013). Descriptive statistics were done for quantitative data as minimum & maximum of the range, median and first & third inter-quartile range, as well as mean ± SD (standard deviation). We calculated the number and percentages for qualitative data. Inferential analyses were done for quantitative variables using the two-tailed paired t-test for two dependent groups with parametric data, and the Wilcoxon signed rank test for two dependent groups with non-parametric data. Correlations were calculated using the two-tailed Pearson correlation for numerical parametric data, the two-tailed Spearman’s rho test for numerical non-parametric and qualitative data, and the two-tailed partial correlation when controlling for a HYRS. The level of significance was taken at *p* value <0.05.

## 3. Results

Twenty patients were included in this study; six were men (30%), and 14 were women (70%). The age of the patients at the time of surgery ranged from 48 to 82 years, with a mean age of 61.1 years. The duration of illness before surgery ranged from five to 30 years, with a mean duration of 11.7 years. The mean ± SD of HYRS of PD stage was 3.6 ± 0.7 with a range from 2.5 to 5.0. No correlation was found between age, sex, family history, or the preoperative associated medical conditions and the one-year follow-up results (Table 1).

The total surgical time of the first-stage surgery ranged from 133 to 280 min, with a mean ± SD of 214.8 ± 44.3 min. The average number of MER tracks used for mapping the STN at a single side was 1.4.

### 3.1. Motor Scores (UPDRS III)

UPDRS III motor score improved by mean ± SD of 20.3 ± 8.9 (49.3%) from the preoperative “off” medication to the one-year follow-up of the “off” medication “on” stimulation score (*p* < 0.001). Tremors improved by mean ± SD of 8.0 ± 5.9 (81.6%) with *p* < 0.001. Rigidity improved by mean ± SD of 3.5 ± 2.1 (50.0%) with *p* < 0.001. Bradykinesia improved by mean ± SD of 5.7 ± 2.7 (39.3%) with *p* < 0.001. The axial score improved by mean ± SD of 2.0 ± 1.2 (38.5%) with *p* < 0.001 (Table 2, Figure 14).

We did not find a correlation between the preoperative improvement in the UPDRS III score, bradykinesia, and axial score after levodopa intake and the one-year follow-up values. A positive correlation was found between the preoperative improvement in tremors score (*r* = 0.563, *p* = 0.010) and rigidity score (*r* = 0.485, *p* = 0.030) with the postoperative tremors and rigidity, respectively.

Regarding the “on” medication state, a total improvement by mean ± SD of 7.0 ± 9.7 (36.5%) from the preoperative “on” medication motor subscore to the one-year follow-up “on” medication “on” stimulation score (*p* < 0.001). Tremors improved by mean ± SD of 2.5 ± 3.8 (78.1%) with *p* = 0.002. Rigidity improved by mean ± SD of 1.6 ± 1.6 (61.5%) with *p* < 0.001. Bradykinesia improved by mean ± SD of 2.7 ± 3.9 (31.4%) with *p* = 0.002. The axial score improved by mean ± SD of 0.3 ± 1.4 (10.7%) with *p* = 0.046 (Table 3).

The “off” medication “off” stimulation score at one year decreased by a mean of 0.4 (1%) compared to the preoperative “off” medication score. This change was found to be non-significant (*p* = 0.452). The “on” medication “off” stimulation at one year increased compared to the preoperative “on” score by a mean of 1.6 (8.3%) with *p* = 0.13.

### 3.2. Activity of Daily Living

The “off” medication UPDRS II improved by mean ± SD of 8.8 ± 4.5 (73.8%) with *p* < 0.001 (Figure 15). The “on” medication UPDRS II improved by mean ± SD of 1.7 ± 5.5 (17.9%), but this improvement did not prove to be statistically significant. The “off” medication SERS improved at the one-year follow-up by mean ± SD of 37.5 ± 15.9 (104.2%) with *p* < 0.001.

### 3.3. Mental State

The UPDRS I score showed non-significant change from a preoperative score of 1.7 ± 1.2 to a one-year score of 1.9 ± 1.3 (Table 4). The MMSE also showed non-significant change (Table 4).

### 3.4. Medications

LEDD decreased in the one-year follow-up by mean ± SD of 849.4 ± 448.1 mg/dL (54.1%) with *p* < 0.001. The UPDRS IVa score improved at the one-year follow-up by mean ± SD of 2.8 ± 2.1 (65.1%) from the preoperative score with *p* < 0.001. The UPDRS IVb score improved at the one-year follow-up by mean ± SD of 1.7 ± 1.1 (48.6%) from the preoperative score with *p* < 0.001 (Table 5, Figure 16).

### 3.5. Complications

All patients reported a postoperative headache that was relieved within a few days. This headache may be attributed to the surgical intervention itself or to the small film of pneumocephalus that appeared on the postoperative CT scans of all the patients. This pneumocephalus was transient and disappeared within days on follow-up CT scans. Three patients complained of memory difficulties; one patient experienced a worsening of his daytime hallucinations, one patient experienced numbness of the scalp at the incision site, and one patient experienced an increase in his depressive symptoms. These adverse events were transient and disappeared within a few weeks. One patient suffered a worsening of speech when turning the stimulation on. One patient developed postoperative right frontal hemorrhage at the site of the cortical penetration by the electrode, and she suffered deterioration of the level of consciousness. The hematoma was evacuated and she achieved complete recovery within one month. One patient suffered from wound dehiscence at the IPG implantation site. The IPG was removed and replaced with a new one in a subcutaneous pocket in the anterior abdominal wall.

## 4. Discussion

In this study we present a snapshot of our experience at UIC in the management of advanced PD using STN-DBS by reporting our preoperative and intraoperative methods as well as the clinical outcomes of 20 patients who were operated on from 2013 to 2014 and followed up for a year. Our experience with DBS started in 2001, and over the years we have modified and improved our techniques [55,56,57].

We have done our preoperative target calculations using a combination of the standard indirect atlas-based X, Y, and Z coordinates of the STN at 12 mm lateral, 3 mm posterior, and 6 mm inferior to the MCP, and direct visualization of the STN on T2 weighted images.

We believe that intra-operative physiological and clinical confirmation of the target is crucial in the final position confirmation. The initial anatomical and radiological planning is also essential in target selection. Accurate preoperative planning would decrease the intra-operative time needed for the MER and the number of the microelectrodes tracks needed to reach the target, and subsequently decrease the complications. This fact is supported by our results, as our average number of MER tracks was 1.4 tracks and our average surgical time was 214.8 min.

Twenty patients were included in the current study; their demographic characteristics are similar to those of the patients included in other studies (Table 6). We only included patients who had the disease for more than five years; this time period is recommended by CAPSIT-PD guidelines to improve the accuracy of diagnosing idiopathic PD, because the defining features of some of the atypical Parkinsonian syndromes may appear late in the course of the disease, e.g., eye movement abnormalities in PSP and autonomic dysfunction in MSA [28,29,30,31]. This time interval is agreed upon by most authors in the literature, as the mean duration of the disease before surgery in previous studies ranged from 6.8 to 16.4 years. The mean ± SD HYRS was 3.6 ± 0.7, ranging from 2.5 to 5.0.

In this study, we report significant improvement in all the motor aspects of PD (UPDRS III) and in ADL (UPDRS II) in the “off” medication state following bilateral STN stimulation. This was accompanied by significant reduction of both medication doses and medication-related adverse effects. This improvement of motor function was significant for the total UPDRS III score and for all the motor subscores (tremor, rigidity, bradykinesia, and the axial score). The change in UPDRS motor scores following surgery in the stimulation “on”, medication “off” condition compared to the baseline medication “off” condition was used as the primary outcome measure in many previous studies (Table 7). The improvement of UPDRS III score ranged from 38.3% to 66.5% [26,27,63,67,68,69,71,73,74,75,76,77,79,80,81] at one-year follow-up; with longer follow-up the beneficial effect of stimulation tends to decrease but scores remain significantly better than preoperative scores [22,26,68,69,71,77,79,80]. The deterioration seems to involve the axial manifestations of PD more than the appendicular motor symptoms. This may be caused by the difference in the pathogenesis of the axial symptoms from the other motor manifestations of PD [82]. The improvement of the UPDRS II “off” score in other studies ranged from 17% to 68.5% and this improvement also tends to decrease with a longer follow-up period [58,59,72,77,83].

The “on” medication, “on” stimulation total UPDRS III score and all the motor subscores showed significant improvement compared to the preoperative “on” medication. Some of the previous studies showed significant improvement at one-year UPDRS III “on” score [60,61], others non-significant improvement [58,62,71], but most studies showed deterioration (increase) of this score at longer follow-up periods [26,32,69,74,76]. The UPDRS II “on” score showed little improvement compared to the preoperative “on” state and it was statistically non-significant. Few studies described significant improvement in this score [71]. Most of the previous studies showed non-significant improvement in the “on” state ADL [58,59,60,61,62,64,65,66]. After longer follow-up, previous studies showed significant deterioration in ADL (increase in score) [26,69,72,74,75]. The less obvious effect of stimulation on the UPDRS III and UPDRS II in the medication “on” state in these studies can be explained by an increase in bradykinesia and the axial sub-scores at the one-year follow-up. The insignificant change of the “off” stimulation UPDRS III in both the “on” and the “off” medication states confirms that the improvement in the “on” stimulation scores is attributed to the DBS effect. Medication doses and medication-induced side effects (dyskinesia and motor fluctuations) were reduced by 65.1% and 48.6%, respectively. These results are in agreement with previous studies (Table 8). The reduction of the LEDD in previous studies ranged from 19.5% to 80.7% (Table 8).

The adverse events encountered in this group of patients were within the accepted complication rate for DBS [84]. While we had one patient with ICH and one with wound dehiscence, both fortunately recovered, one without permanent deficits. While DBS is generally considered a safe procedure, complications can occur and are usually classified into procedure-related, hardware-related, and stimulation–related types [84]. The overall incidence of adverse effects related to surgery is 11%, but it is highly variable between studies [85]. Death is very rare after DBS surgery [62,86]. Procedure-related complications include intracranial hemorrhage (0%–3.9%) and sub-optimal electrode placement (0%–2.5%). Hardware related complications account for 5%–18.5% of the complications; they include lead fracture (0%–5%), lead migration (0%–3.5%), infections, and erosions (1%–9.7%). Stimulation-related complications are the most common problems encountered after DBS surgery, but most of them are reversible or can be avoided by adequate adjustment of the stimulation parameters [69,87,88,89]. They include transient confusion and mood changes. Several studies found that the effect of STN-DBS on overall cognitive functions is non-significant and reversible [27,90,91,92], consistent with the lack of changes from baseline in both the UPDRS I and MMSE scores in our study. A commonly observed complication after DBS is weight gain (8.4%), which can be attributed to the decrease in energy output due to the control of the abnormal movements [84,85,89,93]. Other, reversible, stimulation-related adverse events include contralateral numbness, facial pulling, eye deviation, dysphonia, and speech disturbances. Stimulation may also cause serious psychiatric adverse events such as acute depression but they are reversible upon changing the stimulation parameters. In addition, DBS may induce exacerbation of pre-existing mood disturbances and it is recommended that all patients are screened preoperatively for psychiatric problems [92,94,95]. Nevertheless, patients should not be flat-out denied surgery based on a history of psychiatric illness. In some cases preoperative psychosis or depressive symptoms may be caused by medications, and may therefore improve with dose reduction after surgery [96].

## 5. Conclusions

Despite the limitations of this study as the assessments were not blinded to before vs. after surgery, to medication status, or to DBS status, we can conclude that bilateral deep brain stimulation of the subthalamic nucleus significantly improves motor symptoms, activities of daily living, and medication-induced complications in patients with advanced PD.

## Figures and Tables

**Figure 1 brainsci-06-00058-f001:**
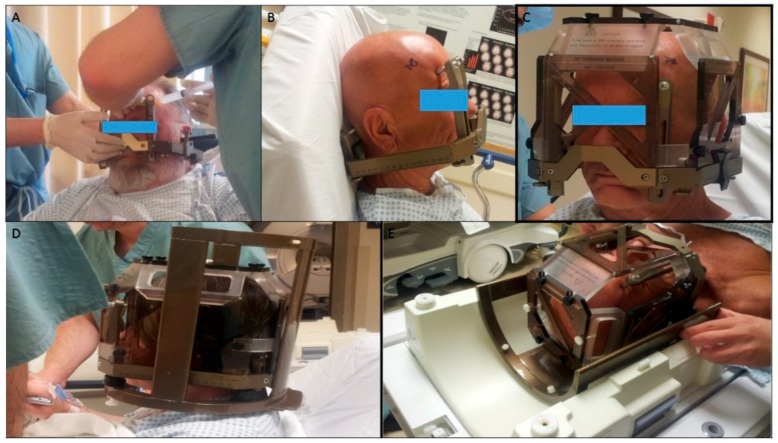
Different steps of Leksell frame application. (**A**) Application of the frame with the ear bars; note that the assistant is holding the frame in position with a lateral bar parallel to the intercommissural line while the senior surgeon is injecting a local anesthetic at the site of pin fixation; (**B**) position of the frame after its application; (**C**) the magnetic resonance imaging (MRI) localizer attached to the frame base; (**D**) the MRI localizer and the table adaptor attached to the frame base; (**E**) the table adaptor fitting to the MRI table.

**Figure 2 brainsci-06-00058-f002:**
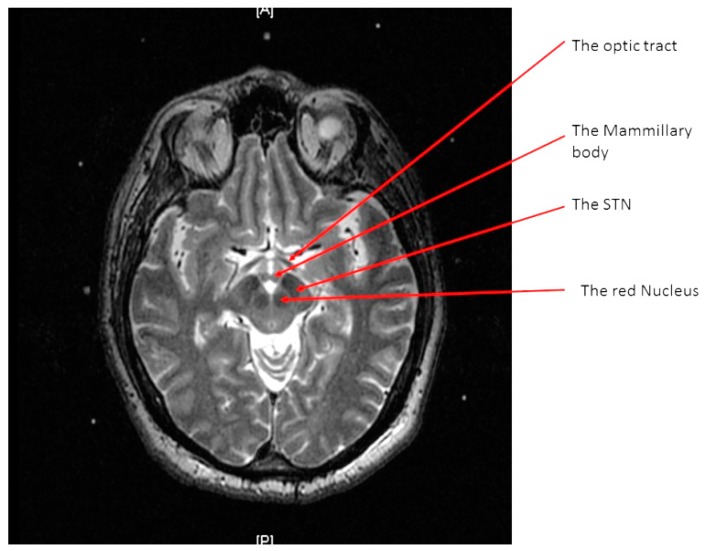
An axial T2 weighted magnetic resonance imaging (MRI) image at the level of the subthalamic nuclei (STN).

**Figure 3 brainsci-06-00058-f003:**
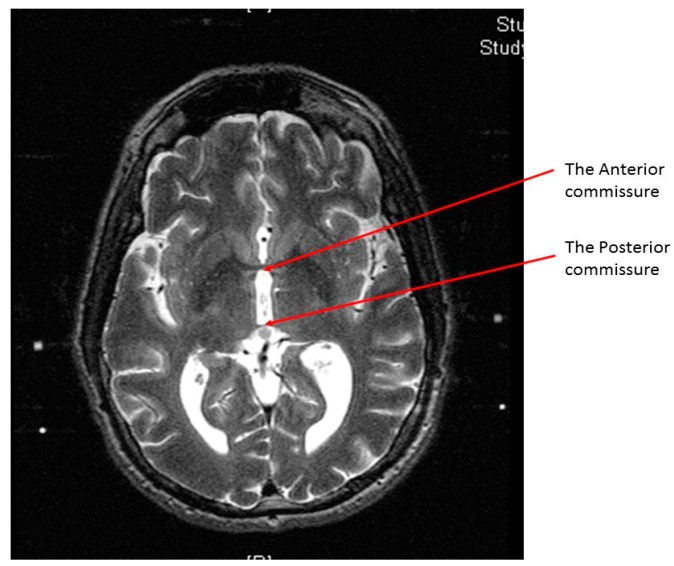
An axial T2 weighted magnetic resonance imaging (MRI) image showing the anterior commissure and the posterior commissure.

**Figure 4 brainsci-06-00058-f004:**
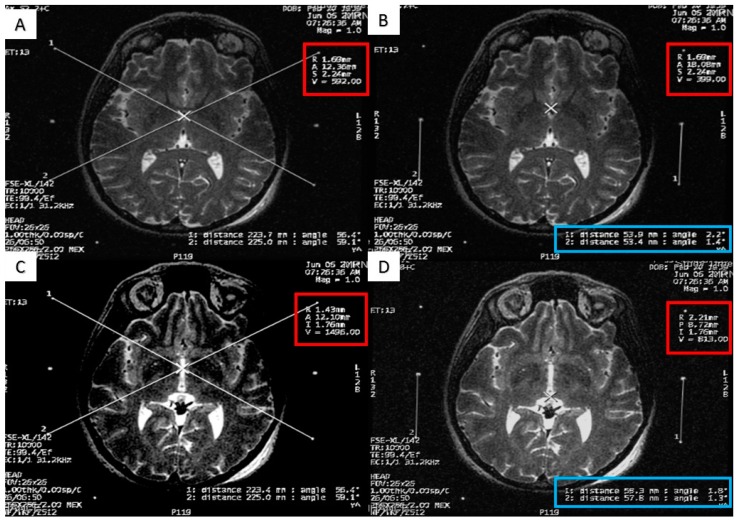
Calculating the anterior commissure (AC) and posterior commissure (PC) coordinates using the magnetic resonance console. (**A**) Two diagonal lines intersecting at the center of the frame at the AC level with the magnetic resonance imaging (MRI) coordinates of the center of the frame shown inside the red square; (**B**) a crosshair at the posterior margin of the AC, with the MRI coordinates of the AC shown inside the red square. Two lines are drawn between the middle and the lower fiducials on both sides of the frame and their lengths (in the blue rectangle) are used to calculate the Z coordinate of the AC. (**C**) Two diagonal lines intersecting at the center of the frame at the PC level with the MRI coordinates of the center of the frame shown inside the red square; (**D**) a crosshair at the anterior margin of the PC, with the MRI coordinates of the PC shown inside the red square. Two lines are drawn between the middle and the lower fiducials on both sides of the frame and their lengths (in the blue rectangle) are used to calculate the Z coordinates of the PC.

**Figure 5 brainsci-06-00058-f005:**
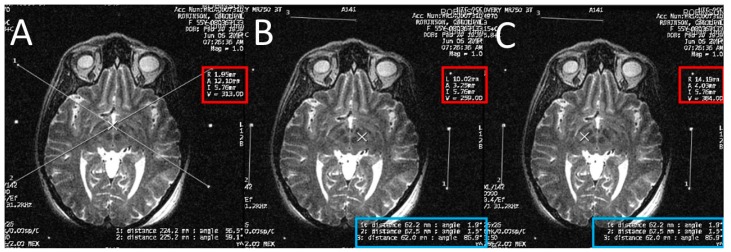
Calculating the subthalamic nucleus (STN) coordinates from the magnetic resonance imaging (MRI) console. (**A**) Two diagonal lines intersecting at the center of the frame at the STN level with MRI coordinates of the center of the frame shown inside the red square; (**B**) a crosshair at the center of the left STN, with its MRI coordinates shown inside the red square; two lines are drawn between the middle and lower fiducials on both sides of the frame and their lengths (in the blue rectangle) are used to calculate the Z coordinate; (**C**) a crosshair at the center of the right STN, with its MRI coordinates shown inside the red square; two line are drawn between the middle and lower fiducials on both sides of the frame and their lengths (in the blue rectangle) are used to calculate the Z coordinate.

**Figure 6 brainsci-06-00058-f006:**
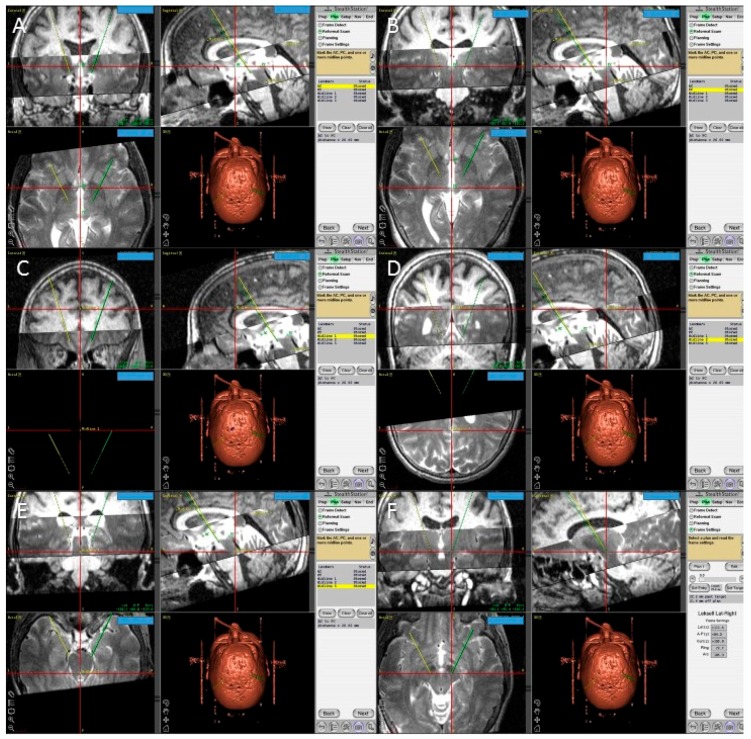
Screen shots from the FrameLink software of the StealthStation showing fused T1 and T2 magnetic resonance imaging (MRI) images of the patient and the planning process with identification of the posterior edge of the anterior commissure (**A**); the anterior edge of the posterior commissure PC (**B**); three midline points (**C**–**E**); and the final coordinates of the right subthalamic nucleus (**F**).

**Figure 7 brainsci-06-00058-f007:**
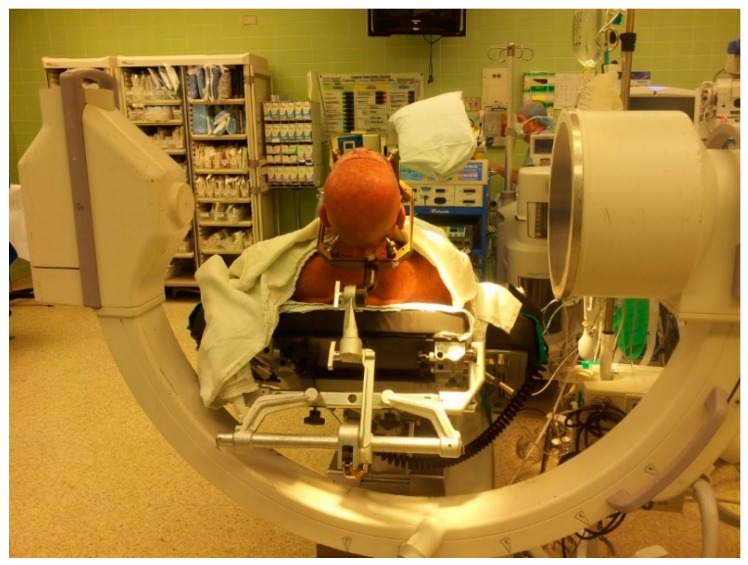
Position of the patient on the operating room table with the Leksell frame fixed to the table through a Mayfield adaptor and the C- arm positioned around the patient.

**Figure 8 brainsci-06-00058-f008:**
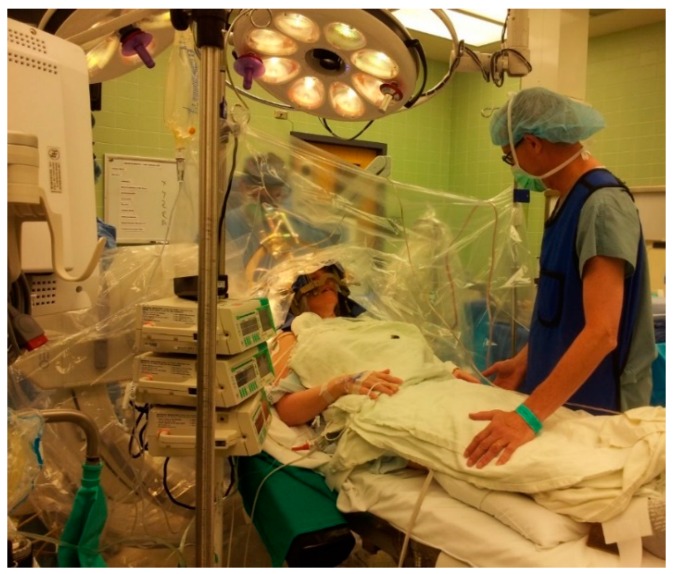
The final position of the patient. Note the transparent draping to allow better communication.

**Figure 9 brainsci-06-00058-f009:**
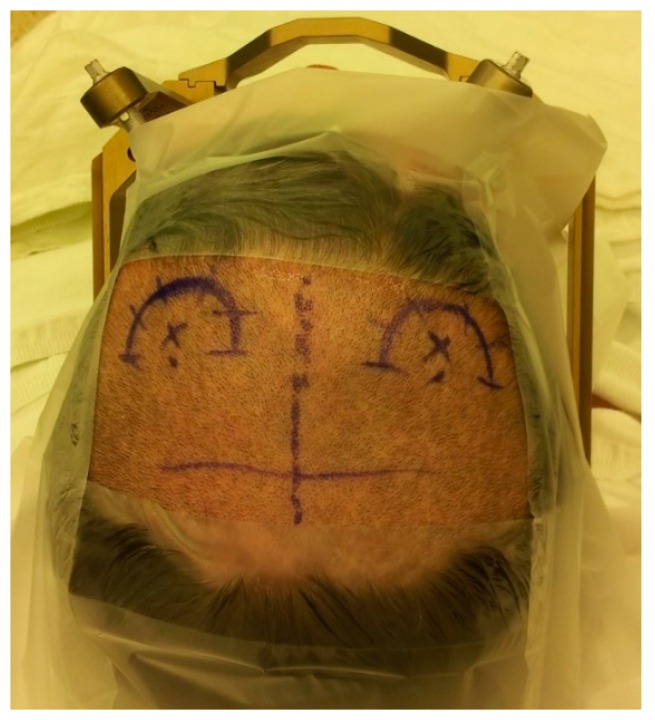
The site of the two semicircular incisions marked on both sides of the midline; with the two burr holes’ positions marked with an X.

**Figure 10 brainsci-06-00058-f010:**
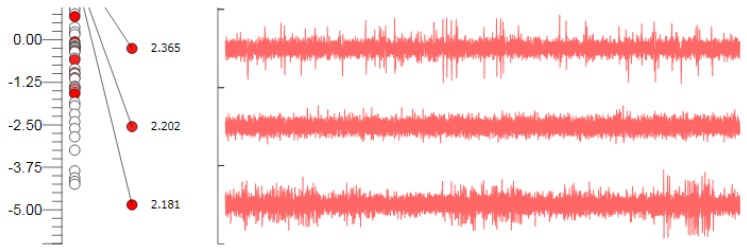
Microelectrode recording appearance of the subthalamic nucleus signal; note the increase in background activity, with high amplitude irregular firing.

**Figure 11 brainsci-06-00058-f011:**
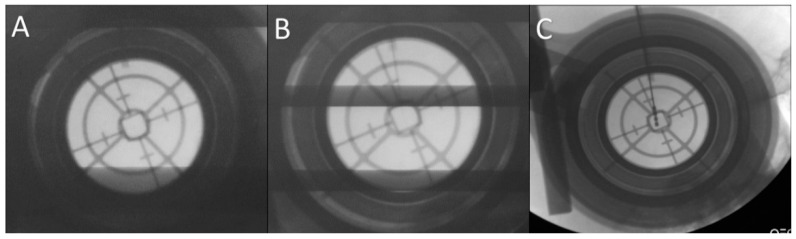
Fluoroscopic confirmation of the target approach. (**A**) Confirmation of the position of the stereotactic cannula; (**B**) the microelectrode is advanced to the target under fluoroscopic guidance; (**C**) the final position of the deep brain stimulation electrode confirmed.

**Figure 12 brainsci-06-00058-f012:**
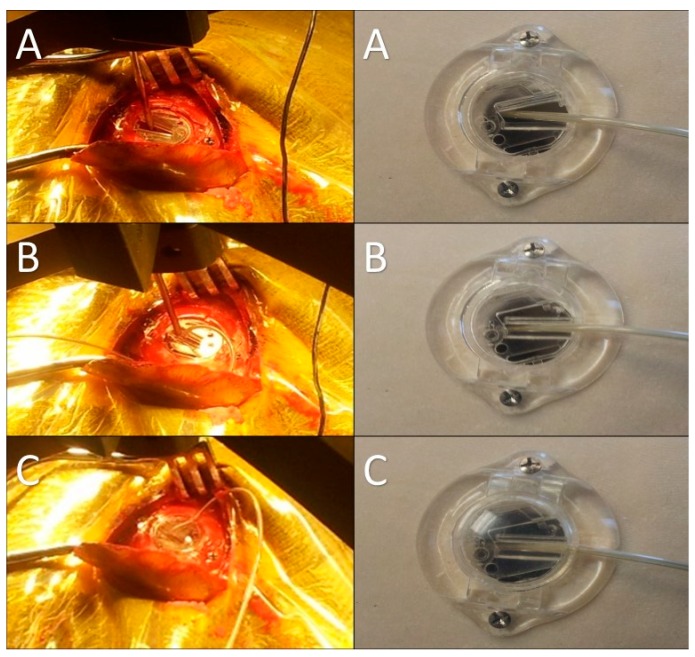
Intraoperative pictures and corresponding model images showing the steps of electrode fixation using the Stimloc device. (**A**) A special locking piece placed onto the Stimloc base with the electrode passing through it; (**B**) the electrode is locked in place with this piece after removing the stylet and moving the electrode out of the cannula; (**C**) the final step of the electrode fixation: the Stimloc cap is placed and fixed over its base.

**Figure 13 brainsci-06-00058-f013:**
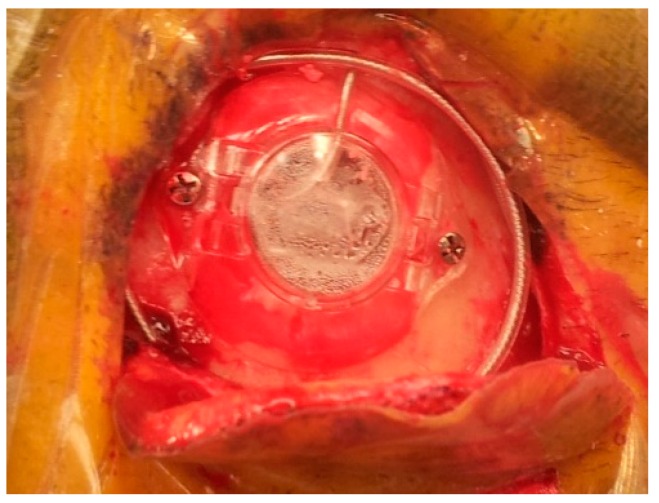
The final appearance of the electrode fixed using the Stimloc device; the excess of the electrode is coiled around the burr hole to create a strain relief loop.

**Figure 14 brainsci-06-00058-f014:**
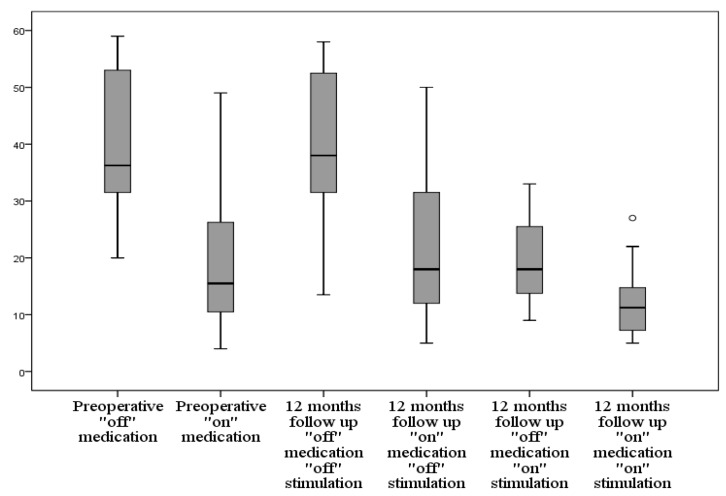
Graph showing the changes of UPDRS III score preoperatively and at 12-month follow-up in different medication and stimulation combinations; UPDRS Unified Parkinson’s Disease Rating Scale.

**Figure 15 brainsci-06-00058-f015:**
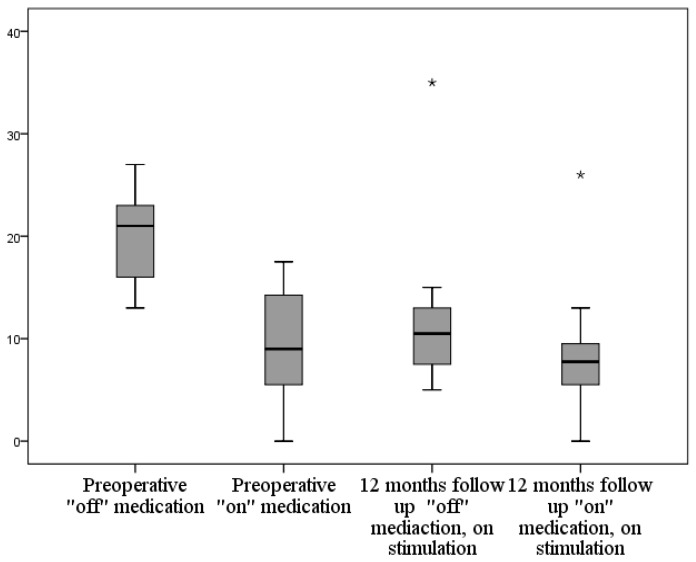
Graph showing the changes of UPDRS II score preoperatively and at 12-month follow-up in “on” and “off” medication conditions; UPDRS Unified Parkinson’s Disease Rating Scale.

**Figure 16 brainsci-06-00058-f016:**
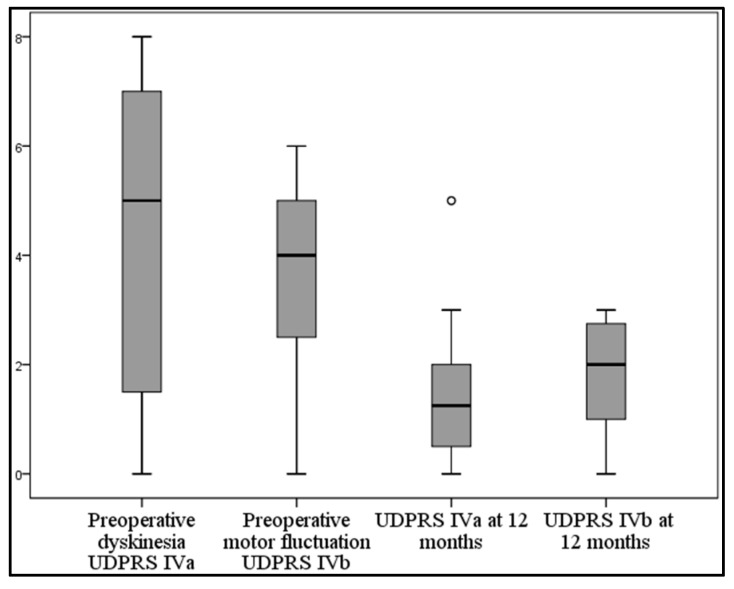
Graph showing the preoperative and 12-month follow-up of UPDRS IVa and IVb; UPDRS Unified Parkinson’s Disease Rating Scale.

**Table 1 brainsci-06-00058-t001:** Preoperative demographic and medical characteristics.

**Variables**		**Mean ± SD**	**Range**
Age (years)		61.1 ± 8.7	48.0–82.0
HYRS		3.6 ± 0.7	2.5–5.0
Duration of illness (years)		11.7 ± 6.4	5.0–30.0
		**Number**	**Percentage (%)**
Sex	Male	6	30.0
	Female	14	70.0
Family history of PD		4	20.0
Smoking		3	15.0
Alcohol intake		4	20.0
Drugs addiction		2	10.0
Hypertension		5	25.0
Coronary artery diseases		1	5.0
Diabetes Mellitus		2	10.0

Total = 20; HYRS Hoehn and Yahr Rating Scale; PD Parkinson’s disease; SD standard deviation; % percentage of change

**Table 2 brainsci-06-00058-t002:** Changes in the “off” medications UPDRS III and UPDRS II scores.

Score (Range)	Measures	Preoperative	One Year	Δ Change	%	^ *p*
UPDRSIII “off” (0–108)	Mean ± SD	40.0 ± 12.4	19.7 ± 7.1	−20.3 ± 8.9	49.3%	<0.001 *
	Median (IQR)	36.3 (31.3–54.0)	18.0 (13.4–26.8)	−19.5 (−26.0–−16.3)		
	Range	20.0–59.0	9.0–33.0	−36.0–2.0		
Tremors “off” (0–28)	Mean ± SD	9.8 ± 6.9	1.8 ± 1.7	−8.0 ± 5.9	81.6%	<0.001 *
	Median (IQR)	10.0 (3.0–16.8)	1.3 (0.0–3.0)	−7.5 (−14.4–−3.0)		
	Range	0.0–19.0	0.0–5.0	−17.0–0.0		
Rigidity “off” (0–20)	Mean ± SD	7.0 ± 3.4	3.5 ± 2.4	−3.5±2.1	50.0%	<0.001 *
	Median (IQR)	8.0 (4.0–9.8)	3.3 (2.0–5.0)	−3.0 (−4.9–−2.0)		
	Range	1.0–12.0	0.0–10.0	−9.0–−1.0		
Bradykinesia “off” (0–32)	Mean ± SD	14.5 ± 4.4	8.8 ± 3.4	−5.7±2.7	39.3%	<0.001 *
	Median (IQR)	14.0 (12.0–17.8)	7.8 (6.3–11.8)	−6.0 (−7.8–−4.5)		
	Range	7.5–23.0	5.0–16.0	−9.5–2.0		
Axial “off” (0–16)	Mean ± SD	5.2 ± 1.3	3.3 ± 1.5	−2.0 ± 1.2	38.5%	<0.001 *
	Median (IQR)	5.3 (4.3–6.0)	3.0 (2.1–3.9)	−2.0 (−2.5–−1.6)		
	Range	3.0–7.0	1.5–8.0	−3.5–2.5		
UPDRSII “off” (0–52)	Mean ± SD	20.1 ± 4.2	11.2 ± 6.4	−8.8 ± 4.5	73.8%	<0.001 *
	Median (IQR)	21.0 (16.0–23.0)	10.5 (7.3–13.0)	−9.5 (−12.0–−8.0)		
	Range	13.0–27.0	5.0–35.0	−13.0–8.0		

Total = 20; Δ Change from before to after (negative values indicate a reduction); *p* probability (*p*-value); SD standard deviation; UPDRS Unified Parkinson’s Disease Rating Scale; % percentage of change; ^ Wilcoxon Signed Ranks Test; * Significant.

**Table 3 brainsci-06-00058-t003:** Changes in the “on” medications UPDRS III and UPDRS II scores.

Score (Range)	Measures	Preoperative	One Year	Δ Change	%	^ *p*
UPDRS III “on” (0–108)	Mean ± SD	19.2 ± 11.7	12.2 ± 5.5	−7.0 ± 9.7	36.5%	<0.001 *
	Median (IQR)	15.5 (10.3–28.1)	11.3 (7.1–15.1)	−4.0 (−14.1–1.6)		
	Range	4.0–49.0	5.0–27.0	−27.0–15.0		
Tremors “on” (0–28)	Mean ± SD	3.2 ± 4.1	0.7 ± 1.3	−2.5 ± 3.8	78.1%	0.002 *
	Median (IQR)	1.5 (0.0–5.8)	0.0 (0.0–1.0)	−1.0 (−2.5–0.0)		
	Range	0.0–14.0	0.0–4.0	−13.0–0.0		
Rigidity “on” (0–20)	Mean ± SD	2.6 ± 2.1	1.0 ± 1.1	−1.6 ± 1.6	61.5 %	<0.001 *
	Median (IQR)	2.0 (1.0–4.5)	1.0 (0.0–1.0)	−1.0 (−2.0–−1.0)		
	Range	0.0–7.0	0.0–4.0	−5.0–0.0		
Bradykinesia “on” (0–32)	Mean ± SD	8.6 ± 4.6	6.0 ± 2.9	−2.7 ± 3.9	31.4%	0.002 *
	Median (IQR)	7.0 (6.0–13.0)	5.0 (4.1–6.8)	−2.0 (−5.5–−0.6)		
	Range	1.5–18.0	1.0–14.0	−9.0–8.0		
Axial “on” (0–16)	Mean ± SD	2.8 ± 1.4	2.5 ± 1.7	−0.3 ± 1.4	10.7%	0.046 *
	Median (IQR)	3.0 (1.6–3.0)	2.0 (1.0–3.0)	−0.3 (−1.0–0.0)		
	Range	1.0–6.0	1.0–8.0	−2.0–5.0		
UPDRSII “on” (0–52)	Mean ± SD	9.5 ± 5.5	7.8 ± 5.4	−1.7 ± 5.5	17.9%	0.073
	Median (IQR)	9.0 (5.3–14.4)	7.8 (5.3–9.5)	−0.3 (−4.9–0.0)		
	Range	0.0–17.5	0.0–26.0	−11.0–16.0		

Total = 20, Δ Change from before to after (negative values indicate a reduction); *p* probability (*p*-value); SD standard deviation; UPDRS Unified Parkinson’s Disease Rating Scale; % percentage of change; ^ Wilcoxon Signed Ranks Test; * Significant.

**Table 4 brainsci-06-00058-t004:** Changes in UPDRS I and MMSE scores.

Score (Range)	Measures	Preoperative	One Year	Δ Change	^ *p*
UPDRS I (2–16)	Mean ± SD	1.7 ± 1.2	1.9 ± 1.3	0.2 ± 0.7	^ 0.059
	Median (IQR)	2.0 (1.0–2.8)	2.0 (1.0–3.0)	0.0 (0.0–0.8)	
	Range	0.0–4.0	0.0–4.0	0.0–2.0	
MMSE (0–30)	Mean ± SD	29.2±1.0	28.9 ± 1.3	−0.3 ± 1.1	# 0.249
	Median (IQR)	29.0 (29.0–30.0)	29.0 (28.0–30.0)	0.0 (−1.0–0.0)	
	Range	27.0–30.0	25.0–30.0	−4.0–2.0	

Total = 20, Δ Change from before to after (negative values indicate a reduction); IQR interquartile range; *p* probability (*p*-value); MMSE Mini Mental Status Exam; SD standard deviation; UPDRS Unified Parkinson’s Disease Rating Scale; ^ Wilcoxon Signed Ranks Test, # Paired *t*-test.

**Table 5 brainsci-06-00058-t005:** Changes in the LEDD and UPDRS IV score.

Score	Measures	Preoperative	One Year	Δ Change	%	^ *p*
LEDD (mg/dL)	Mean ± SD	1570.8 ± 662.9	721.4 ± 312.4	−849.4 ± 448.1	54.1%	<0.001 *
	Median (IQR)	1675.0 (832.5–2175.0)	691.0 (416.5–975.0)	−800.0 (−1287.5–−360.0)		
	Range	616.0–2500.0	300.0–1200.0	−1514.0–−270.0		
UPDRS IVa	Mean ± SD	4.3 ± 2.9	1.6 ± 1.5	−2.8 ± 2.1	65.1%	<0.001 *
	Median (IQR)	5.0 (1.3–7.0)	1.3 (0.3–2.0)	−3.0 (−4.9–−1.0)		
	Range	0.0–8.0	0.0–5.0	−6.0–1.0		
UPDRS IVb	Mean ± SD	3.5 ± 1.8	1.8 ± 1.0	−1.7 ± 1.1	48.6%	<0.001 *
	Median (IQR)	4.0 (2.3–5.0)	2.0 (1.0–2.9)	−2.0 (−2.0–−1.0)		
	Range	0.0–6.0	0.0–3.0	−4.0–0.0		

Total = 20, Δ Change from before to after (negative values indicate a reduction, which is an indicator of improvement); LEDD levodopa equivalent daily dose; *p* probability (*p*-value); SD standard deviation; UPDRS Unified Parkinson’s Disease Rating Scale; % percentage of change; ^ Wilcoxon Signed Ranks Test; * Significant.

**Table 6 brainsci-06-00058-t006:** Demographic characteristics of patients in previous studies of bilateral subthalamic nucleus stimulation in PD.

Study	*N*	F/U (Months)	Mean Age at Surgery (Years)	Mean Disease Duration before Surgery (Years)	Mean “off” HYRS
Limousin et al. [58]	20	12	56 ± 8	14 ± 5	4.6 ± 0.5
Ostergaard et al. [59]	26	12	59 ± 8 (30–75)	15 ± 5	3.7
Simuni et al. [60]	12	12	58 ± 11	12 ± 4	3.8
Thobois et al. [61]	14	12	56.9 ± 6	13.5 ± 4.4	3.6
Vesper et al. [62]	38	12	55.6 (47–72)	13	3.5
Vingerhoets et al. [63]	20	21 ± 8	63 ± 8	16 ± 5	*
Volkmann et al. [64]	16	12	60.2 ± 9.8	13.1 ± 5.9	*
Herzog et al. [65]	20	24	60 ± 6	15 ± 5	3.7
Krause et al. [66]	24	29.8 ± 8.5	57.7	14.4 ± 5.8	4.3
Fraix et al. [67]	24	12	55.7 ± 7.3	16.3 ± 5.4	4.5
Kleiner-Fisman et al. [68]	25	24	57.2 ± 11.7	13.4 ± 4.3	*
Krack et al. [69]	42	60	55 (34–68)	14.6 ± 5	*
Romito et al. [70]	33	25.7 ± 13.5	56.8 ± 7.1	13.8 ± 5.5	*
Visser-Vandewalle et al. [71]	20	48	60.9 ± 8.1	15.0 ± 4.4	*
Rodrguez-Oroz et al. [72]	10	48	62.0 (53–73)	13 (4–23)	*
Rodrguez-Oroz et al. [73]	49	48	*	*	*
Wider et al. [74]	37	60	64.9 ± 7.6	14.4 ± 4.9	*
Gervais-Bernard et al. [75]	23	48	55.1 ± 7.2	12.9 ± 3.2	*
Castrioto et al. [26]	18	120	52.9 ± 7.9	13.4 ± 4.8	*
Weaver et al. [76]	70	36	60.7 ± 8.9	11.3 ± 4.7	3.3 ± 0.8
Li et al. [77]	195	60	58.2 ± 10	6.8 (5–15)	*
Tabbal et al. [78]	72	6	48.4 ± 9.8 (28–69)	14.5 ± 6.5 (4–29)	*
This study	20	12	61.1±8.7 (48–82)	11.7 ± 6.4 (5–30)	3.6 ± 0.7 (2.5–5)

*N*: number of patients; F/U: duration of follow-up; HYRS Hoehn and Yahr Rating Scale; PD: Parkinson’s disease; UPDRS: Unified Parkinson’s Disease Rating Scale; *: no available data

**Table 7 brainsci-06-00058-t007:** Motor and activity of daily living outcomes of patients in previous studies of bilateral subthalamic nucleus stimulation in PD.

Study	% of Change of UPDRS III “off”	% of Change of UPDRS III “on”	% of Change of UPDRS II “off”
Limousin et al. [58]	−60%	−10%	−60%
Ostergaard et al. [59]	−64.3%	*	−66.3%
Simuni et al. [62]	−47.1%	−2%	*
Thobois et al. [64]	−55%	*	−52.8%
Vesper et al. [60]	−48%	−39%	−38%
Vingerhoets et al. [63]	−45%	*	−37%
Volkmann et al. [65]	−60.3%	*	−55.9%
Herzog et al. [61]	−57.2%	−35%	−43.6%
Krause et al. [66]	−38.3%	*	−17%
Fraix et al. [67]	−66.5	−16%	*
Kleiner-Fisman et al. [68]	−39%	*	−26%
Krack et al. [69]	−54%	+47.6	−49%
Romito et al. [70]	−51.6%	*	−68.5%
Visser-Vandewalle et al. [71]	−42.8%	−22.6%	−59.4%
Rodrguez-Oroz et al. [72]	−62.0%	*	−62.0%
Rodrguez-Oroz et al. [73]	−50%	−12.3%	−43.1%
Wider et al. [74]	−25.5%	+26%	*
Gervais-Bernard et al. [75]	−55%	−11.2%	−38.1%
Castrioto et al. [26]	−25.3%	+44.7	−23%
Weaver et al. [76]	−30.1%	+11.1%	−6.25%
Li et al. [77]	−60.3%	*	−54.2%
Tabbal et al. [78]	−47%	*	*
This study	−49.3%	−36.5%	−73.8%

%: percentage, *: no available data, UPDRS: Unified Parkinson’s Disease Rating Scale, PD: Parkinson’s disease.

**Table 8 brainsci-06-00058-t008:** Change of LEDD and medication-induced complications in patients in previous studies of bilateral subthalamic nucleus stimulation in PD.

Study	% of Change of LEDD	% of Change of Dyskinesia	% of Change of Motor Fluctuation
Limousin et al. [58]	*	−60%	*
Ostergaard et al. [59]	−19.5%	−86%	−79%
Simuni et al. [62]	−55%	−64.3%	−32.4%
Thobois et al. [64]	−65.5%	−76%	*
Vesper et al. [60]	−53%	−71.9%	−35%
Vingerhoets et al. [63]	−79%	−92%	−95%
Volkmann et al. [65]	−65.3%	−83.3%	*
Herzog et al. [61]	−67%	−85%	*
Krause et al. [66]	−30%	−70%	−16%
Fraix et al. [67]	−80.7%	−86.7%	*
Kleiner-Fisman et al. [68]	−42%	−65.5%	−58%
Krack et al. [69]	−63%	−65%	*
Romito et al. [70]	−56.2%	−83.9%	−94.2%
Visser-Vandewalle et al. [71]	−47.2%	−79%	−78.4%
Rodrguez-Oroz et al. [72]	−50.0%	−53%	*
Rodrguez-Oroz et al. [73]	−35%	−59%	*
Wider et al. [74]	−56.9%	−85.4%	−84.2%
Gervais-Bernard et al. [75]	−54.4%	−60%	*
Castrioto et al. [26]	−36.3%	−68.8%	−46.9%
Weaver et al. [76]	−35.7%	−75%	*
Tabbal et al. [78]	45%	*	*
This study	−54.1%	−65.1%	−48.6%

LEDD levodopa equivalent daily dose; PD: Parkinson’s disease, UPDRS: Unified Parkinson’s Disease Rating Scale, %: percentage, *: no available data.

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
