# Peer review of "Improvement of Advanced Parkinson’s Disease Manifestations with Deep Brain Stimulation of the Subthalamic Nucleus: A Single Institution Experience"

_brainsci, 2016, doi:10.3390/brainsci6040058_

Round 1

Reviewer 1 Report

The authors are to be commended for collecting and reporting their outcomes for DBS therapy applied to people with advanced Parkinson’s disease.  This center serves a very large patient population and reporting outcomes is important and helpful to the Parkinson’s disease community.

Author Response

We want to thank the esteemed reviewer for kind assessment of our work.

Reviewer 2 Report

The authors describe the clinical and surgical methods, and key results, from 20 PD patients treated with STN DBS at their institution. The details and results are important additions to the data on STN DBS, though not novel.

Clarifying some key points, and addressing a few errors, would improve the manuscript.

The manuscript can be understood as it stands, but it needs English language proofreading throughout, most notably to correct superfluous or missing articles (for instance, the 3rd word in the title).

Some key information is missing or unclear.

Line 73: “the data of 20 patients diagnosed with advanced PD who underwent bilateral STN-DBS at the UIC in the period from 2013 to 2014”: which 20 patients? I.e. are these all the patients who had bilateral STN DBS at UIC in 2013-2014, or were any such patients excluded from the report?

Section on neuropsychological and psychiatric evaluation starting at line 121:  This section needs clarification and I believe also perpetuates some incorrect conclusions.

“Exclude any patient with … psychiatric illness”: “psychiatric illness” is not defined (do you include personality disorders? off-period anxiety? ADHD in childhood?). Besides, it cannot reflect your actual practice. For example, one patient had pre-existing psychosis (line 358) and another had pre-existing depressive symptoms (lines 359-360).

Line 147: TE value is missing

Line 153: I don’t see the spreadsheet. It can be uploaded as supplementary material or on FigShare or similar.

Lines 285, 300: “no positive correlation was found”: this is an odd way of wording it. Was there a negative correlation? How about just give (r = x.xx, p = 0.yy) here?

Line 313: “statistically increased” -- not sure what that means but there was no significant increase.

What part of the electrode was aimed at the target? The tip? The bottom edge of contact 0? Etc.

Some enthusiasm needs to be toned down, and some assertions are far from universally held.

Line 61:  STN DBS “proved to be superior to best medical therapy” for PD:  this is true in highly selected patients.

Line 106: “After five or more years of dopaminergic therapy, patients typically begin to experience motor fluctuations and dyskinesias; this stage of the disease is called “Advanced PD”. Those are the candidates for DBS.”: That last sentence needs revision. For many patients with 5 or even 15 years of PD, DBS is not indicated (or, at least, experts will disagree).

Line 108 “In late phase of PD, patients become unresponsive to levodopa.” This assertion is factually incorrect.

Next sentence: “Those patients are not considered surgical candidates for DBS.” I do not agree. Often these patients—i.e., those for whom levodopa management no longer provides adequate benefit—are ideal candidates for STN DBS.

Lines 264-266: how long did you wait after giving LD before doing the off stim on meds rating? And how long after LD were the on stim on meds ratings?

Line 434: ‘The insignificant change of the “off” stimulation UDPRS III in both the “on” and the “off” medication states, confirms that … there was no significant progression of the PD symptoms in the follow up period.’ -- Can you remove the last part of this? (“there was no significant progression of the PD symptoms in the follow up period”) At best you can say that the change in signs measured under these controlled conditions was not statistically significant, and there’s no point saying that.

Line 464ff: “Stimulation may also cause serious psychiatric adverse events as transient acute depression, and manic episodes that is why all patients should be well assessed preoperatively for psychiatric problems.”  This sentence is probably true, but in the context of line 122 and following, the authors here suggest that patients with prior psychiatric problems should be excluded from surgery. I strongly disagree and have elsewhere explained why (Parkinsonism Relat Disord 2007; 13(8):546. PMID 17303465   DOI: 10.1016/j.parkreldis.2006.12.007).

Around line 472, the conclusions should be qualified by recognizing the primary limitation of this study: the assessments were not blind to before vs after surgery, to medication status, or to DBS status.

Less crucial points follow.

The introduction could benefit from a nod to work with animal models, e.g. Aziz 1991, or early subthalamotomy reports e.g. PubMed ID # 9010390 or 11215596.

Line 155: please spell out MCP on its first use

Line 294: sentence fragment

Table 8: consider adding PubMed ref. PMID: 17876242

Ref 37 author name needs correction

Author Response

Dear Editor,

We greatly appreciate a thorough review of our submitted work and constructive comments of the reviewers. Below is the detailed response to the reviewers’ concerns. All corrections and changes have been incorporated in the revised manuscript.

Reviewer 1:

Comments to the Author
-           The authors describe the clinical and surgical methods, and key results, from 20 PD patients treated with STN DBS at their institution. The details and results are important additions to the data on STN DBS, though not novel.

Thank you for your kind comments!

-       Clarifying some key points, and addressing a few errors, would improve the manuscript.

The manuscript can be understood as it stands, but it needs English language proofreading throughout, most notably to correct superfluous or missing articles (for instance, the 3rd word in the title).

We corrected the title according to your comment.

-           Some key information is missing or unclear. Line 73: “the data of 20 patients diagnosed with advanced PD who underwent bilateral STN-DBS at the UIC in the period from 2013 to 2014”: which 20 patients? I.e. are these all the patients who had bilateral STN DBS at UIC in 2013-2014, or were any such patients excluded from the report?

These were 20 consecutive patients who underwent bilateral STN DBS for advanced Parkinson disease and who completed 1-year follow up evaluation during the fellowship term of the primary author. The text was corrected to reflect this.

-                Section on neuropsychological and psychiatric evaluation starting at line 121:  This section needs clarification and I believe also perpetuates some incorrect conclusions. “Exclude any patient with … psychiatric illness”: “psychiatric illness” is not defined (do you include personality disorders? off-period anxiety? ADHD in childhood?). Besides, it cannot reflect your actual practice. For example, one patient had pre-existing psychosis (line 358) and another had pre-existing depressive symptoms (lines 359-360).

The section on neuropsychological and psychiatric evaluation was modified and made clearer to follow your valuable advice.

-                Line 147: TE value is missing

We added TE value

-                Line 153: I don’t see the spreadsheet. It can be uploaded as supplementary material or on FigShare or similar.

We eliminated the mention of our spreadsheet as it does not carry educational value

-                Lines 285, 300: “no positive correlation was found”: this is an odd way of wording it. Was there a negative correlation? How about just give (r = x.xx, p = 0.yy) here? Line 313: “statistically increased” -- not sure what that means but there was no significant increase.

The text was corrected based on reviewer’s recommendations.

-                What part of the electrode was aimed at the target? The tip? The bottom edge of contact 0? Etc.

We generally placed the deepest electrode contact (#0) at or just beyond the target location.

-                Some enthusiasm needs to be toned down, and some assertions are far from universally held. Line 61:  STN DBS “proved to be superior to best medical therapy” for PD:  this is true in highly selected patients.

We based our statement on published results of Weaver et al., 2009

-                Line 106: “After five or more years of dopaminergic therapy, patients typically begin to experience motor fluctuations and dyskinesias; this stage of the disease is called “Advanced PD”. Those are the candidates for DBS.”: That last sentence needs revision. For many patients with 5 or even 15 years of PD, DBS is not indicated (or, at least, experts will disagree).

The sentence regarding the duration after which symptoms become refractory to medications was modified.

-                Line 108 “In late phase of PD, patients become unresponsive to levodopa.” This assertion is factually incorrect.

The sentence on levodopa responsiveness was modified.

-                Next sentence: “Those patients are not considered surgical candidates for DBS.” I do not agree. Often these patients—i.e., those for whom levodopa management no longer provides adequate benefit—are ideal candidates for STN DBS.

We respect the reviewer’s opinion and agree that there are indeed some exceptions to general rules. Normally, however, we consider ideal candidates to DBS those who start suffering motor fluctuation on levodopa, or those who suffer severe dyskinesia or other disabling levodopa complications. Ideally, to get benefit from surgery patient should at least show 30% improvement on UDPRS motor sub score after levodopa challenge. Based on multiple published studies, it appears that those who get no benefit at all from levodopa will not benefit from DBS.

-                Lines 264-266: how long did you wait after giving LD before doing the off stim on meds rating? And how long after LD were the on stim on meds ratings?

There was no fixed duration we waited for to rate the patient’s condition. Basically, we gave the patient levodopa and then waited until he/she felt that the best on state was reached. After assessing the patient in the “on medication” state, we turned on the stimulation and reassessed him in the “on medication / on stimulation” state.

-                Line 434: ‘The insignificant change of the “off” stimulation UDPRS III in both the “on” and the “off” medication states, confirms that … there was no significant progression of the PD symptoms in the follow up period.’ -- Can you remove the last part of this? (“there was no significant progression of the PD symptoms in the follow up period”) At best you can say that the change in signs measured under these controlled conditions was not statistically significant, and there’s no point saying that.

The sentence was removed as per reviewer’s recommendation.

-                Line 464ff: “Stimulation may also cause serious psychiatric adverse events as transient acute depression, and manic episodes that is why all patients should be well assessed preoperatively for psychiatric problems.”  This sentence is probably true, but in the context of line 122 and following, the authors here suggest that patients with prior psychiatric problems should be excluded from surgery. I strongly disagree and have elsewhere explained why (Parkinsonism Relat Disord 2007; 13(8):546. PMID 17303465   DOI: 10.1016/j.parkreldis.2006.12.007).

We concur with the reviewer’s valuable point – we did indeed exclude patients with severe uncontrolled psychiatric illnesses including depression. We added a reference in support of this approach.

-                Around line 472, the conclusions should be qualified by recognizing the primary limitation of this study: the assessments were not blind to before vs after surgery, to medication status, or to DBS status.

We changed the text based on the reviewer’s comment.

-                Less crucial points follow. The introduction could benefit from a nod to work with animal models, e.g. Aziz 1991, or early subthalamotomy reports e.g. PubMed ID # 9010390 or 11215596.

We added the recommended references – thank you!

-                Line 155: please spell out MCP on its first use. Line 294: sentence fragment

Done.

-                Table 8: consider adding PubMed ref. PMID: 17876242

We added the recommended reference – thank you!

-                Ref 37 author name needs correction

We corrected the author name in the mentioned reference.
